# Nonlinear Responses of Phytoplankton Communities to Environmental Drivers in a Tourist-Impacted Coastal Zone: A GAMs-Based Study of Beihai Silver Beach

**DOI:** 10.3390/biology15010034

**Published:** 2025-12-25

**Authors:** Dewei Cheng, Xuyang Chen, Yun Chen, Fangchao Zhu, Ying Qiao, Li Zhang, Ersha Dang

**Affiliations:** 1Key Laboratory of Tropical Marine Ecosystem and Bioresource, Fourth Institute of Oceanography, Ministry of Natural Resources, Beihai 536015, China; chengdewei@4io.org.cn (D.C.); chenxuyang@4io.org.cn (X.C.); zhufangchao@4io.org.cn (F.Z.); qiaoying@4io.org.cn (Y.Q.); 2Guangxi Laboratory of Oceanography (GXLO), Beihai 536015, China; 3Marine Environmental Engineering Center, South China Sea Institute of Oceanology, Chinese Academy of Sciences, Guangzhou 510301, China; 4Ecological Environment Monitoring and Scientific Research Center, Taihu Basin & East China Sea Ecological Environment Supervision and Administration Bureau, Ministry of Ecology and Environment, Shanghai 200125, China; chenyun@thdhjg.mee.gov.cn

**Keywords:** phytoplankton community, Beihai Silver Beach, GAMs, environmental factor, species richness, diversity index

## Abstract

A three-year study in Beihai Silver Beach revealed that phytoplankton communities were consistently dominated by diatoms, with a major bloom of *Skeletonema costatum* occurring in autumn 2021. Species richness peaked in summer, while abundance hotspots were found near the Fengjia River estuary. Using advanced statistical models, phosphorus was identified as the primary driver, showing complex nonlinear relationships with both species richness and diversity. Surprisingly, lead concentration showed a positive correlation with species richness, revealing complex stressor interactions. These findings highlight the importance of phosphorus control and demonstrate the value of advanced modeling for coastal ecosystem management.

## 1. Introduction

Coastal waters play a pivotal role in the marine economy and serve as an epicenter where the interactive impacts of human activities, environmental changes, and climate change are most directly and intensely felt [1,2]. As the primary producers in marine ecosystems, phytoplankton are highly sensitive to environmental stressors, which makes their community structure a reliable indicator of ecosystem health [3]. Extensive research has established that the species composition, diversity, and succession of dominant phytoplankton taxa are strongly influenced by multiple environmental variables, including water temperature, nutrient concentrations, and pollutants such as emerging contaminants [4,5]. Given these sensitivities, phytoplankton communities have become vital biological indicators for assessing water quality and ecological integrity, particularly in anthropogenically impacted coastal regions [6,7].

Beihai Yintan, located on the northern coast of the South China Sea in Guangxi, China, is a renowned coastal tourism destination, often called “China’s First Beach”. In recent years, intensive coastal development, tourism, and terrestrial inputs have led to escalating environmental challenges, including eutrophication, water quality deterioration, and increasing pollution loads [8]. While previous studies have provided preliminary data on hydrochemical parameters and sediment properties in this area, systematic research on the phytoplankton community structure is notably lacking [9,10]. Specifically, there is a critical gap in understanding the mechanistic responses of phytoplankton to multiple interacting environmental drivers. Moreover, the prevailing reliance on conventional correlation analyses in existing research fails to capture the complex nonlinear relationships between environmental factors and phytoplankton community attributes, thereby limiting predictive accuracy and the formulation of effective management strategies [11].

To address these knowledge gaps, a comprehensive multi-seasonal survey was conducted in the nearshore waters of Beihai Yintan between 2020 and 2022. This study aims to elucidate the phytoplankton community composition, succession patterns of dominant species, and spatiotemporal variations in biodiversity. By integrating traditional correlation analysis with advanced generalized additive models (GAMs), which are particularly suited for capturing nonlinear ecological responses [12]. We seek to identify key environmental drivers governing community dynamics and to disentangle the nonlinear relationships between environmental variables and phytoplankton indicators. Additionally, the cumulative ecological impacts of multiple environmental stressors are evaluated. This research not only addresses a critical knowledge gap in the phytoplankton ecology of this region, but also offers a scientific foundation for ecological monitoring, pollution mitigation, and adaptive management of coastal waters, thereby enhancing the understanding of ecological response mechanisms in highly utilized coastal tourist environments.

## 2. Materials and Methods

### 2.1. Study Area

Beihai Silver Beach is located in the southern Guangxi Zhuang Autonomous Region, China, within the southeastern sector of Beihai City. Its geographic coordinates extend from 21°23′ to 21°29′ N and 109°02′ to 109°08′ E. The area spans approximately 24 km of coastline, bounded by Daguansha to the east and Guan Tou Ling to the west. Adjacent to the urban center of Beihai to the northwest and bordering the Beibu Gulf to the south, Silver Beach constitutes a key region for urban expansion and coastal tourism. It is recognized as a highly significant coastal resort area along China’s southern seaboard.

Surface water samples were obtained from 15 stations within the nearshore environment of Silver Beach (Figure 1), with one sample collected per station for a total of 15 samples. The sampling design focused on the central tourism district and the Fengjia River estuary to encompass regions subject to substantial anthropogenic pressure. Seasonal sampling was conducted once per quarter in December 2020, September 2021, and August 2022, with each campaign completed within two consecutive days, representing characteristic hydrological conditions during winter, autumn, and summer, respectively.

### 2.2. Sampling and Analysis

Phytoplankton samples were collected using a shallow-water Type III plankton net (inner mouth diameter: 37 cm, mesh size: 0.077 mm). This approach may undersample smaller microplankton and nanoplankton, including many single-celled diatoms, small dinoflagellates, and other pico-sized fractions. Consequently, the community composition and species richness likely reflect a bias towards larger-sized taxa, and the total phytoplankton diversity may be underestimated. Our conclusions are therefore interpreted primarily within the context of the “net-phytoplankton” community. At each station, vertical tows were performed three times from the bottom to the surface. The initial and final readings of the flowmeter were recorded. Samples were preserved with Lugol’s solution. After sedimentation for at least 48 h, the samples were concentrated to a suitable volume. A 0.1 mL aliquot of the homogenized concentrate was placed in a counting chamber, and species identification and enumeration were conducted under an OLYMPUS CX22 optical microscope. For colonial or chain-forming species, counts represent the number of individual cells. All samples were stored in the dark and subsequently transported to the laboratory for taxonomic analysis.

Taxonomic identification was primarily based on the following references: Atlas of Marine Biology of China (Vol. 1) [13], Atlas of Common Marine Planktonic Diatoms in Chinese Waters [14], Marine Planktonic Diatoms in China [15], Freshwater Algae of China [16], Marine Benthic Diatoms in China (Vol. 1) [17] and Marine Benthic Diatoms in China (Vol. 2) [18].

At each station, in situ environmental parameters including temperature (T), salinity (S), dissolved oxygen (DO), and pH were measured synchronously using a YSI6000 multiparameter water quality probe. Water samples were collected with a 2.5 L Niskin bottle at 0.5 m below the surface and 2 m above the seabed to obtain surface and bottom layers, respectively. After thorough mixing, subsamples were stored in 500 mL polyethylene bottles, kept at 4 °C in a dark icebox, and transported to the laboratory within 24 h.

For nutrient analysis, water samples were filtered through GF/F glass fiber filters (47 mm diameter). The filtrate was used for the determination of nitrite (NO_2_^−^-N), ammonium (NH_4_^+^-N), nitrate (NO_3_^−^-N), dissolved inorganic phosphate (DIP), silicate (SiO_3_^2−^), total nitrogen (TN), and total phosphorus (TP). Petroleum hydrocarbons were extracted using n-hexane and quantified by ultraviolet spectrophotometry. Chemical oxygen demand (COD) was measured using the alkaline potassium permanganate method.

All sampling, preservation, and analytical procedures were strictly followed the national standards of China: Specifications for Oceanographic Survey and Marine Monitoring—Part three: Sample Collection, Storage and Transportation.

### 2.3. Statistical Analysis

#### 2.3.1. Communities Diversity and Dominance

The formulas for calculating dominance (*Y*), the Shannon–Weiner diversity index (*H′*), and the Pielou evenness index (*J*) are as follows [19,20,21]:$$Y=\frac{n_{i}}{N}\cdot f_{i}$$
where *nᵢ* represents the abundance of the *ith* species; *N* denotes the total abundance of all species in the study area; *fᵢ* indicates the frequency of occurrence of the *i* species across all sampling stations. A species is considered dominant when *Y* ≥ 0.02.$$H^{\prime }=-\sum_{i=1}^{S}P_{i}\log_{2}P_{i}\;J=\frac{H^{\prime }}{\log_{2}S}$$
where *S* indicates the total number of species in the sample, and *Pᵢ* signifies the ratio of the abundance of the *i* species to the total number of individuals (*nᵢ*/*N*).

#### 2.3.2. GAMs Analysis

Generalized Additive Models (GAMs) were employed to conduct an in-depth analysis of the nonlinear relationships between multiple environmental factors and phytoplankton′ s response variables (*S* and *H′*). The GAMs were implemented using the “mgcv” package in R, with square-root transformed phytoplankton biodiversity parameters as response variables and effectively screened environmental factors (after removing collinear variables) as predictors [22,23]. The model is expressed as follows:(1)$$g\left(\mu \right)=\beta +\sum_{i=1}^{n}f_{i}\left(X_{i}\right)+\varepsilon$$

In Equation (1), g(μ) represents the link function of the response variable, *β* denotes the intercept term, n indicates the number of predictor variables (X*_i_*), and f*_i_* refers to the natural spline smooth function of each predictor, describing the nonlinear relationship between the transformed mean response variable and the predictors [24]. The residual error term ε accounts for all unexplained variance not captured by the predictors [25].

The smooth curves generated by the model not only reveal the strength of the influence of environmental factors on phytoplankton biodiversity parameters but also reflect the direction of these effects. This allows for the identification of the most significant predictors [26,27], thereby providing critical insights into the ecological responses of phytoplankton biodiversity under varying environmental conditions.

Model fit was evaluated using parameters such as degrees of freedom, F-value, *p*-value, and coefficient of determination (R^2^). Degrees of freedom help determine whether the relationship between environmental factors and phytoplankton community characteristics is nonlinear, while the F-value indicates the magnitude of influence exerted by environmental factors on these community metrics. The proportion of total variation in the response variable explained by the model was assessed using R^2^. All GAM fitting and validation procedures were performed using R version 4.4.3.

## 3. Results

### 3.1. Characteristics of Water Quality Indicators

The results of environmental factor investigations are summarized in Table 1, revealing distinct seasonal variations across different parameters. Salinity, dissolved oxygen, and active phosphate exhibited minimal fluctuations across the three sampling campaigns. In contrast, suspended particulate matter reached its highest level in August 2022 (summer), with comparable values observed in the other two surveys. Active silicate concentrations peaked in December 2020 (winter), followed by August 2022 (summer), and were lowest in September 2021 (autumn). Nitrite, nitrate, and chlorophyll-a levels were highest in August 2022 (summer), intermediate in December 2020 (winter), and lowest in September 2021 (autumn). Ammonium and chemical oxygen demand showed maximum values in December 2022 (winter), moderate levels in September 2021 (autumn), and minimum values in August 2022 (summer). Total phosphorus was elevated in September 2021 (autumn) with negligible variations between the other two surveys, while total nitrogen was highest in December 2020 (winter), followed by August 2022 (summer), and lowest in September 2021 (autumn) (Table 1).

### 3.2. Variation Characteristics of Phytoplankton Community Structure

#### 3.2.1. Species Composition of Phytoplankton

In winter 2020, a total of 72 phytoplankton species belonging to 2 phyla were identified in the surveyed marine area. Diatoms were the most dominant group, with 65 species accounting for 90.28% of the total, followed by dinoflagellates comprising 7 species (9.72%). The number of phytoplankton species per sampling station during winter ranged from 12 to 27 (Figure 2).

In autumn 2021, 113 phytoplankton species from 3 phyla were recorded. Diatoms remained predominant, represented by 102 species (90.27%), while dinoflagellates accounted for 10 species (8.85%), and cyanobacteria constituted 1 species (0.88%). The species count per station in summer varied between 35 and 68 (Figure 2).

In summer 2022, 111 species spanning 5 phyla were identified. Diatoms continued to dominate with 90 species (81.08%), followed by dinoflagellates (13 species, 11.71%), chlorophytes (4 species, 3.60%), cyanobacteria (3 species, 2.70%), and a single species of chromophytic flagellate (0.90%). The number of species per station in spring ranged from 26 to 49 (Figure 2). Overall, phytoplankton species richness exhibited a distinct seasonal pattern: autumn > summer > winter.

#### 3.2.2. Dominant Species

The dominant phytoplankton species comprised 7 taxa in winter 2020, 5 in autumn 2021, and 9 in summer 2022. Diatoms predominated across all surveys. *Skeletonema costatum* was identified as a dominant species in all three sampling campaigns, with dominance values of 0.380 (winter 2020), 0.395 (autumn 2021), and 0.023 (summer 2022). Species such as *Chaetoceros lorenzianus*, *Chaetoceros affinis*, and *Bacteriastrum hyalinum* occurred as dominants in two different seasons. In contrast, only one dinoflagellate species, *Peridinium pentagonum*, was recorded as a dominant species, primarily during the summer 2022 survey, with a dominance value of 0.023 (Table 2).

#### 3.2.3. Community Composition Characteristics

In December 2020, the phytoplankton cell abundance in the surveyed area ranged from 0.65 × 10^4^ to 18.00 × 10^4^ cells/L, with an average value of 3.33 × 10^4^ cells/L. Diatoms were the most abundant group, accounting for 98.8% of the total with a mean abundance of 3.29 × 10^4^ cells/L, followed by dinoflagellates at 0.04 × 10^4^ cells/L, representing 1.2% of the total. In September 2021, phytoplankton cell abundance varied from 427.80 × 10^4^ to 7509.58 × 10^4^ cells/L, averaging 3193.08 × 10^4^ cells/L. Diatoms remained the dominant group, comprising 99.8% of the total with a mean abundance of 3187.65 × 10^4^ cells/L. Dinoflagellates followed with a mean of 4.46 × 10^4^ cells/L (1.2%), while cyanobacteria were the least abundant group at 0.97 × 10^4^ cells/L, accounting for 0.03%. During the summer of 2022, phytoplankton cell abundance ranged from 67.13 × 10^4^ to 969.98 × 10^4^ cells/L, with an average of 290.33 × 10^4^ cells/L. Diatoms again constituted the majority (98.8%), with a mean abundance of 272.19 × 10^4^ cells/L. Dinoflagellates followed at 11.13 × 10^4^ cells/L (1.2%). Other groups, including cryptophytes, chlorophytes, and cyanobacteria, were present in lower abundances, with mean values of 1.25 × 10^4^ cells/L, 4.51 × 10^4^ cells/L, and 1.26 × 10^4^ cells/L, respectively.

In terms of horizontal distribution, phytoplankton abundance was relatively uniform across all stations in both winter 2020 and autumn 2021. Abundance in winter 2020 was generally low, with all stations recording values below 10 × 10^4^ cells/L. In contrast, abundance in autumn 2021 was considerably higher, peaking at station T11 with a maximum value of 7509.58 × 10^4^ cells/L. During summer 2022, the spatial distribution of phytoplankton abundance was more heterogeneous. Stations T12 and T15 exhibited higher abundances, exceeding 500 × 10^4^ cells/m^3^, whereas stations T6, T8, and T9 showed lower abundances, all below 100 × 10^4^ cells/L (Figure 3 and Figure 4).

#### 3.2.4. Diversity Index

In December 2020, the phytoplankton diversity index ranged from 1.71 to 4.09, with a mean value of 3.15, while the evenness index ranged from 0.46 to 0.93, averaging 0.77. In September 2021, the diversity index varied between 1.78 and 3.54, with a mean of 2.63, and the evenness index ranged from 0.30 to 0.65, averaging 0.46. In August 2022, the diversity index values were recorded between 2.43 and 4.30, with a mean of 3.62, and the evenness index ranged from 0.51 to 0.82, averaging 0.71 (Figure 5).

To evaluate seasonal variations in the Shannon–Wiener diversity index (H′) and Pielou’s evenness index (J), the non-parametric Kruskal–Wallis test was employed due to the non-normally distributed data. Marked overall differences were detected among the three seasons (Winter 2020, Autumn 2021, Summer 2022) for both indices H′ (χ^2^ = 18.290, df = 2, *p* < 0.05) and J (χ^2^ = 21.831, df = 2, *p* < 0.05).

### 3.3. GAMs-Based Analysis of Phytoplankton Community Characteristic Indices and Environmental Drivers

To further investigate the ecological responses of different phytoplankton community characteristics under various environmental conditions, Generalized Additive Models (GAMs) were constructed. Initially, the Variance Inflation Factor (VIF) test was employed to eliminate variables with high multicollinearity (Figure 6, VIF < 5). In conjunction with outlier detection, 7 and 6 environmental factors were ultimately selected to construct the GAMs for species richness and biodiversity index, respectively. The model equations are as follows:
(2)$$\sqrt{S}\sim s({{\mathrm{PO}}_{4}}^{3-})+s(\mathrm{Pb})+s(\mathrm{TP})+s(\mathrm{DO})+s(\mathrm{Hg})+s(\mathrm{WD})+s(\mathrm{oil})+4.777$$
(3)$$\sqrt{H^{\prime }}\sim s({{\mathrm{PO}}_{4}}^{3-})+s(\mathrm{DO})+s(\mathrm{Pb})+s(\mathrm{oil})+s(\mathrm{WD})+s(\mathrm{Hg})+3.142$$

In Equations (2) and (3), PO_4_^3−^ indicates the concentration of inorganic phosphorus (reactive phosphate), Pb represents the concentration of lead, TP represents the concentration of total phosphorus, DO represents dissolved oxygen, Hg represents the concentration of total mercury, WD represents water depth, and oil represents petroleum. The estimated coefficients of the intercept terms for the S and H′ models are 4.777 and 3.142, respectively.

Table 3 provides information on the robustness of the fitted models. From Table 3, it can be observed that the R^2^ for the S fitted model is 0.91. The value of Generalized Cross-Validation (GCV) is 0.9, indicating that the model has an appropriate level of complexity, fits the data well, and does not overfit the data [26]. For the H′ fitted model, the intercept term is statistically significant, with an R^2^ of 0.436. The GCV value is 0.354, indicating that the model fits the data well and does not overfit the data.

Table 4 and Table 5 present the results of the smooth terms for all predictor variables, with each effective degree of freedom (edf) serving as a measure of the degree of nonlinearity of the curve [27]. As can be seen from the tables, the phytoplankton biodiversity parameters/indicators are determined by the combined effects of multiple environmental factors. For the response variable phytoplankton species richness (S, Table 4), the edf for reactive phosphate (PO_4_^3−^) is 5.333, the reference degrees of freedom (Ref.df) is 6.077, and the *p*-value is less than 9.08 × 10^−4^; the edf for lead concentration (Pb) is 5.534, Ref.df is 6.198, and the *p*-value is 0.00355; the edf for total phosphorus (TP) is 5.326, Ref.df is 6.086, and the *p*-value is 0.00594. Dissolved oxygen (DO), mercury concentration (Hg), water depth (WD), and petroleum (oil) did not show a significant causal relationship with the response variable S in the coastal waters of Beihai Silver Beach during the 2021–2022 period. Other water quality parameters showed a significant causal relationship with the response variable S, and PO_4_^3−^ was the most important driver of S changes in the coastal waters of Beihai Silver Beach during the 2021–2022 period.

When the response variable was the phytoplankton biodiversity index H’ (Table 5), the edf for reactive phosphate (PO_4_^3−^) was 1.195, Ref.df was 1.35, and the *p*-value was 0.0355; the edf for dissolved oxygen (DO) was 5.574, Ref.df was 6.567, and the *p*-value was 0.0269. Lead concentration (Pb), petroleum (oil), water depth (WD), and mercury concentration (Hg) did not show a significant causal relationship with the response variable H′ in the coastal waters of Beihai Silver Beach during the 2021–2022 period. PO_4_^3−^ was the most important driver of H′ changes in the coastal waters of Beihai Silver Beach during the 2021–2022 period.

The effect estimates of each predictor variable on the response variables are shown in Figure 7 and Figure 8. For the response variable phytoplankton species richness (S, Figure 7), the smooth function of reactive phosphate (PO_4_^3−^) shows an increasing trend, and lead concentration (Pb) shows a significant increasing trend. The smooth function of total phosphorus (TP) also shows an increasing trend. The smooth functions corresponding to dissolved oxygen (DO), mercury (Hg), water depth (WD), and oil, denoted as s(DO, 3.24), s(Hg, 1.69), s(WD,1), and s(oil,1), show neither increasing nor decreasing trends. This indicates that during the 2021–2022 period, these four parameters could not explain the variability of S in the coastal waters of Beihai Silver Beach. Through the analysis of the Generalized Additive Model (GAM), it was found that during the 2021–2022 period, PO_4_^3−^ had the most significant impact on the species richness of phytoplankton in the coastal waters of Beihai Silver Beach, followed by lead concentration and total phosphorus. The model failed to establish any strong association between the response variable S and Hg, water depth, or oil.

For the response variable H′ (Figure 8), the effect curves of PO_4_^3−^ and DO show clear nonlinear relationships, with H′ increasing first and then decreasing as PO_4_^3−^ and DO increase. This is consistent with the significance of these two variables in the model. The effect curve of Pb also shows a certain nonlinear trend, but does not reach a statistically significant level. For oil, WD, and Hg, the effect curves are relatively flat and do not show clear nonlinear relationships, which is consistent with the nonsignificance of these variables in the model. In particular, the effect curves of WD and Hg are almost horizontal, indicating that these two variables have little or no impact on H′ (Figure 7 and Figure 8).

## 4. Discussion

This study provides a comprehensive analysis of the phytoplankton community structure and its environmental drivers in the nearshore waters of Beihai Silver Beach, a highly utilized coastal tourist destination. The results reveal distinct spatiotemporal patterns in phytoplankton composition, abundance, and diversity, and further elucidate the complex, often nonlinear, relationships between community metrics and key environmental factors using advanced statistical modeling.

### 4.1. Spatiotemporal Dynamics of Phytoplankton Community

The phytoplankton community in Beihai Silver Beach was consistently dominated by diatoms across all surveyed seasons, which is a typical characteristic of coastal marine ecosystems [1]. However, our multi-seasonal data revealed a nuanced succession pattern. Species richness was highest in summer (111 species from 5 phyla), followed closely by autumn (113 species from 3 phyla), and lowest in winter (72 species from 2 phyla). This pattern aligns with the general understanding that warmer temperatures and increased light availability in summer favor phytoplankton growth and diversity [28]. The low species richness in winter 2020 likely reflects the suboptimal growth conditions associated with lower water temperatures. Notably, while surface water temperatures were similar in autumn 2021 (30.4 °C) and summer 2022 (30.7 °C), the distinct community structures observed suggest that factors beyond temperature, such as nutrient dynamics and pollutant loads, were primary drivers of inter-seasonal differences.

The pronounced spatial heterogeneity in phytoplankton abundance during summer 2022, with notably higher cell densities at stations T12 and T15, suggests localized influences. These stations are situated near the Fengjia River estuary, implying that terrestrial inputs, potentially rich in nutrients, may be a significant driver of phytoplankton biomass in specific areas. This finding underscores the importance of considering point sources of anthropogenic influence in coastal management [29].

The overwhelming dominance of *Skeletonema costatum* across all seasons, particularly in 2021, is noteworthy. The formation of such monospecific blooms can significantly alter trophic dynamics and is often linked to eutrophic conditions and specific hydrographic features [30]. The high abundance in autumn 2021, orders of magnitude greater than in other seasons, indicates the occurrence of a substantial bloom event. This interpretation is further supported by the characteristically low concentrations of silicate (0.09 mg/L) and nitrate (0.005 mg/L) in autumn 2021, which are indicative of a post-diatom bloom environment where these essential nutrients have been heavily depleted. The bloom was likely triggered by a confluence of favorable environmental conditions, such as optimal nutrient ratios and water column stability prior to nutrient drawdown.

A particularly intriguing pattern emerged in autumn 2021: species richness (113 species) was comparable to summer 2022 (111 species), yet chlorophyll-a concentration was exceptionally low (0.01 mg/L) while cell abundance was approximately three orders of magnitude greater than in winter 2020. This decoupling of biomass (chlorophyll-a) from cell abundance and taxonomic richness could be attributed to several factors: (1) the dominance of small-sized species like *S. costatum*, which have a lower chlorophyll content per cell; (2) the likely presence of a high proportion of senescent or stressed cells following the bloom peak, contributing to microscopic counts but not to viable photosynthetic biomass; and (3) the methodological bias associated with the 77 µm mesh net, which may have influenced the perceived relationship between counted cells and integrated chlorophyll measurements.

The Shannon–Weiner diversity and Pielou evenness indices exhibited significant seasonal fluctuations, as confirmed by Kruskal–Wallis tests which revealed statistically significant overall differences across seasons for both indices. The lowest values were recorded in autumn 2021. This statistically confirmed decline in diversity, concurrent with the peak in total abundance and the extreme dominance of *S*. *costatum*, is a classic manifestation of the paradox of enrichment, where increased nutrient loading can lead to competitive exclusion and reduced community evenness [31]. In contrast, the higher and more stable diversity indices in summer 2022 and winter 2020 suggest a more balanced and resilient community structure during those periods, which corresponded with the absence of such extreme monospecific blooms.

### 4.2. Key Environmental Drivers Identified by GAMs

The application of Generalized Additive Models (GAMs) provided deeper insights beyond traditional correlation analyses, effectively capturing the nonlinear responses of phytoplankton community characteristics to environmental stressors. For phytoplankton species richness (S), reactive phosphate (PO_4_^3−^), lead (Pb) concentration, and total phosphorus (TP) were identified as the most significant drivers. The positive, albeit complex, relationships between phosphorus nutrients (PO_4_^3−^ and TP) and species richness are intuitive, as phosphorus is a key limiting nutrient for phytoplankton growth in many marine systems [32]. However, the GAMs revealed that these relationships were nonlinear, suggesting potential threshold effects where the influence of phosphorus on promoting species richness may saturate or change at higher concentrations.

The significant association with lead (Pb), a toxic heavy metal, was unexpected. The model indicated a positive relationship with S, which is counterintuitive given the known toxicity of lead to phytoplankton [33]. This paradoxical result may be attributed to co-varying environmental factors not fully captured in the model, or it may indicate a complex indirect effect, perhaps mediated by the suppression of grazers that are more sensitive to lead than certain phytoplankton species [34]. This highlights the intricate and sometimes counterintuitive nature of ecosystem responses to multiple stressors [35].

Regarding the phytoplankton biodiversity index (H′), the models identified reactive phosphate (PO_4_^3−^) and dissolved oxygen (DO) as the primary significant drivers, both exhibiting clear nonlinear relationships. The dome-shaped response of H′ to increasing PO_4_^3−^ concentration is ecologically significant. It suggests that low to moderate levels of phosphorus can enhance biodiversity, potentially by alleviating resource competition, but beyond a certain threshold, further enrichment leads to a decline in diversity [36,37]. This decline is likely due to the competitive advantage conferred to a few opportunistic species (like *S. costatum*), leading to dominance and a subsequent drop in evenness, as observed in our data [29,38].

The significant nonlinear relationship with DO is also critical. While high DO is generally indicative of a healthy system, the relationship with biodiversity is not straightforward [39]. The dome-shaped curve could reflect the fact that supersaturated DO levels often result from high photosynthetic activity during dense blooms, which are typically characterized by low diversity [40]. Conversely, low DO can indicate respiratory stress or degradation processes, which also negatively impact diversity [41]. Therefore, the highest biodiversity is maintained at moderate DO levels, representing a balanced state without extreme bloom conditions or oxygen deficiency.

Furthermore, our data revealed a striking anomaly in trace metal concentrations during summer 2022, with Cu and Zn levels an order of magnitude higher, and chromium Cr an order of magnitude lower, compared to other seasons. This distinct metallic signature likely points to a specific, transient anthropogenic source during that period, such as anti-fouling paints, localized runoff, or industrial discharge. Given the known toxicity of elevated Cu and Zn to phytoplankton physiology, this anomalous metal cocktail could have acted as an additional stressor, potentially contributing to the unique community structure observed in summer 2022 and underscoring the variable nature of pollutant pressures in this coastal zone.

### 4.3. Implications for Coastal Management

The findings of this study have direct implications for the environmental management of Beihai Silver Beach. The identification of phosphorus (in the form of PO_4_^3−^) as a critical driver for both species richness and biodiversity underscores the need for targeted nutrient management strategies in the watershed [42]. Controlling terrestrial inputs of phosphorus from agricultural runoff, domestic sewage, and other anthropogenic sources should be a priority to mitigate the risk of extreme algal blooms and the associated loss of biodiversity. Furthermore, the significant influence of toxic metals like lead, even if its role is complex, warrants continued monitoring and investigation of heavy metal pollution in the area [35]. The use of advanced modeling approaches like GAMs has proven valuable in diagnosing ecosystem health and identifying potential tipping points or nonlinear responses to environmental change [43]. This approach should be integrated into long-term monitoring programs to improve predictive capacity and support adaptive management strategies for this and similar coastal tourist destinations facing increasing anthropogenic pressure [44]. In conclusion, the phytoplankton community in Beihai Silver Beach exhibits dynamic seasonal and spatial variations, driven by a combination of natural hydrographic conditions and anthropogenic influences. The dominance of diatoms, the occurrence of seasonal blooms, and the fluctuations in diversity are mediated by a complex interplay of environmental factors. The application of GAMs has been instrumental in revealing that nutrients (particularly phosphorus) and dissolved oxygen are primary drivers with nonlinear effects on community structure. These insights are crucial for developing science-based management practices aimed at preserving the ecological integrity and long-term health of this valuable coastal ecosystem. Future research should focus on long-term time-series data and incorporate metabolic and molecular techniques to further unravel the mechanistic links between environmental stressors and phytoplankton responses.

### 4.4. Limitations

While this study offers significant insights, several limitations should be acknowledged. First, the phytoplankton data are based on net tows (77 µm mesh), which undersample pico- and nanoplankton; our conclusions pertain primarily to the net-phytoplankton community. Second, the sampling design, with one campaign per season over three years, captures seasonal snapshots but cannot resolve higher-frequency variability or long-term trends; a continuous time-series would strengthen temporal inferences. Third, the GAMs identify significant associations but cannot definitively establish causal mechanisms. The paradoxical positive relationship with Pb, for example, suggests the influence of unmeasured confounding variables or complex indirect pathways. Finally, the model’s explanatory power for the diversity index H′ (R^2^ = 0.436) was moderate, indicating other important drivers, potentially biological interactions like grazing or viral lysis, were not included.

## 5. Conclusions

In conclusion, the phytoplankton community in Beihai Silver Beach exhibits dynamic seasonal and spatial variations, driven by a combination of natural hydrographic conditions and anthropogenic influences. The dominance of diatoms, the occurrence of seasonal blooms, and the fluctuations in diversity are mediated by a complex interplay of environmental factors. The application of GAMs has been instrumental in revealing that nutrients (particularly phosphorus) and dissolved oxygen are primary drivers with nonlinear effects on community structure. These insights are crucial for developing science-based management practices aimed at preserving the ecological integrity and long-term health of this valuable coastal ecosystem. Future research should focus on long-term time-series data and incorporate metabolic and molecular techniques to further unravel the mechanistic links between environmental stressors and phytoplankton responses.

## Figures and Tables

**Figure 1 biology-15-00034-f001:**
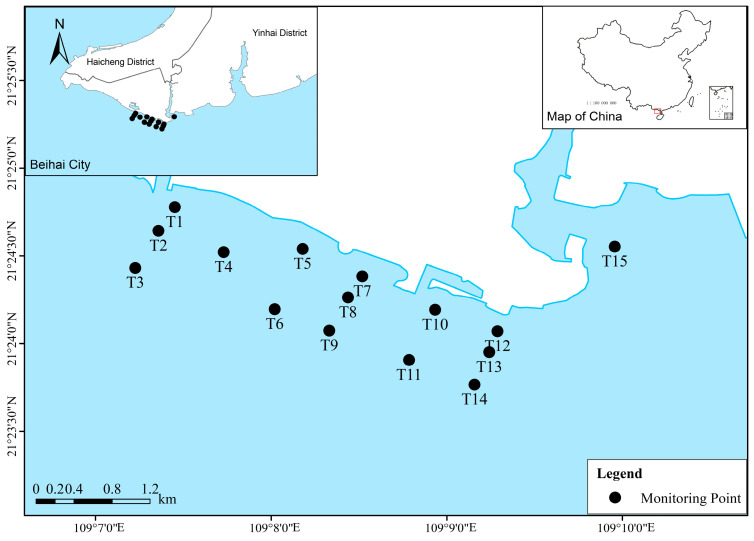
Locations of the sampling sites in the Beihai Silver Beach.

**Figure 2 biology-15-00034-f002:**
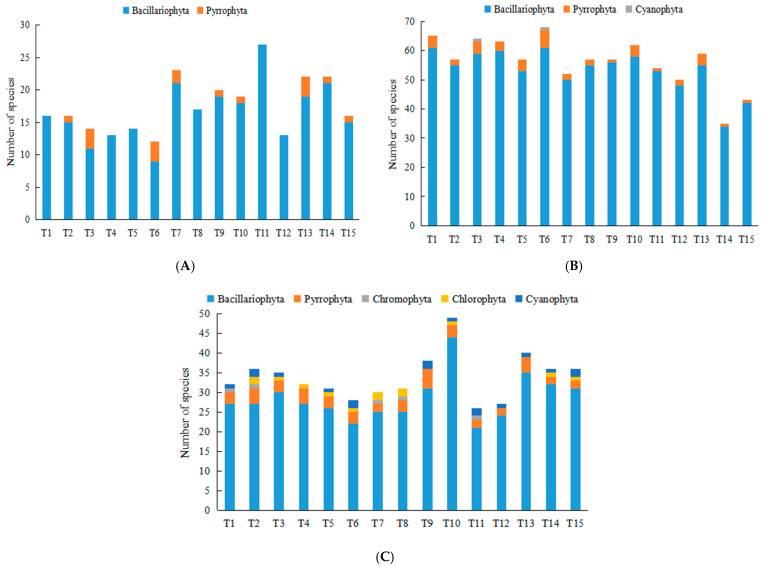
Species composition of phytoplankton at each station in the offshore waters of Beihai Silver Beach ((**A**). December 2020; (**B**). September 2021; (**C**). August 2022).

**Figure 3 biology-15-00034-f003:**
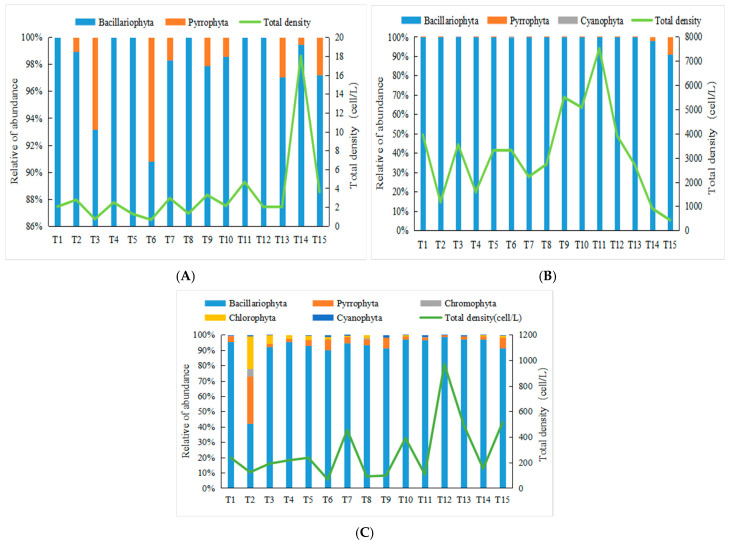
Relative abundance and total density of phytoplankton ((**A**). December 2020; (**B**). September 2021; (**C**). August 2022).

**Figure 4 biology-15-00034-f004:**
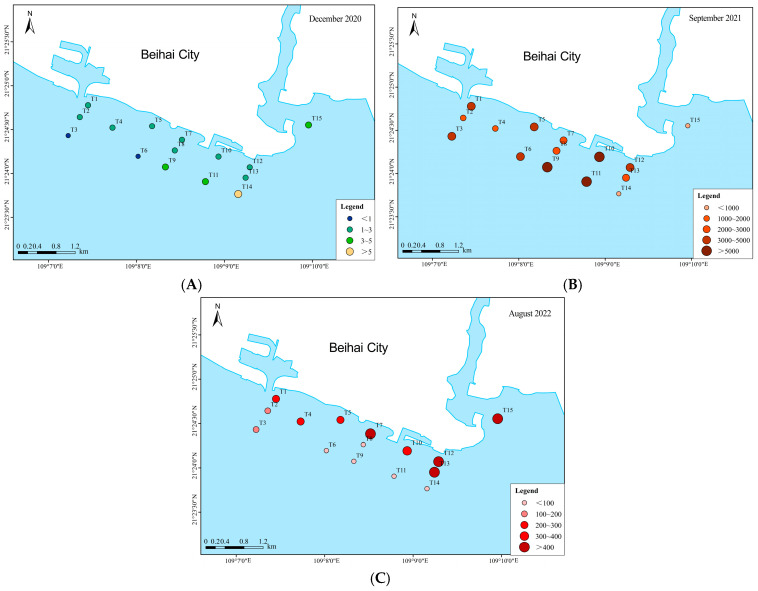
Spatial distribution of phytoplankton abundance ((**A**). December 2020; (**B**). September 2021; (**C**). August 2022).

**Figure 5 biology-15-00034-f005:**
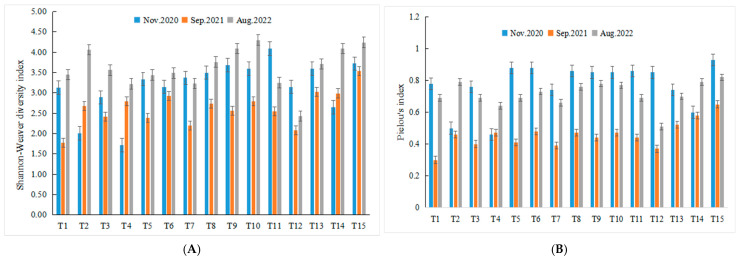
Seasonal change in the diversity of phytoplankton ((**A**). Shannon-Weaver diversity index; (**B**). Pielou’s index).

**Figure 6 biology-15-00034-f006:**
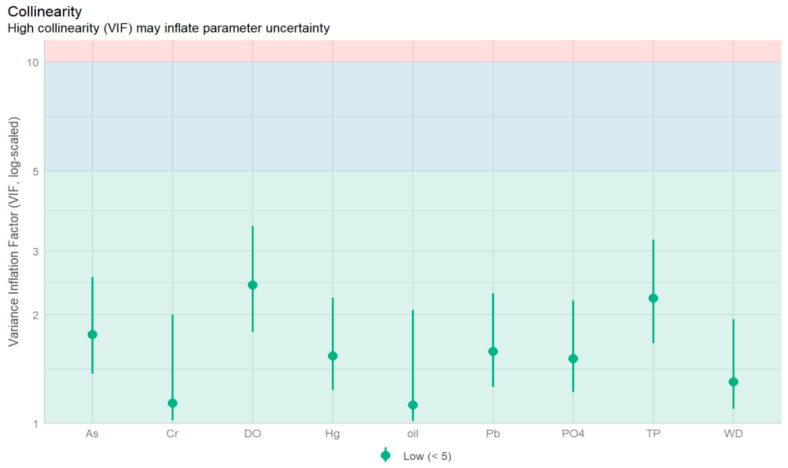
Collinearity Test of Environmental Factors in the Coastal Waters of Beihai Silver Beach (VIF < 5).

**Figure 7 biology-15-00034-f007:**
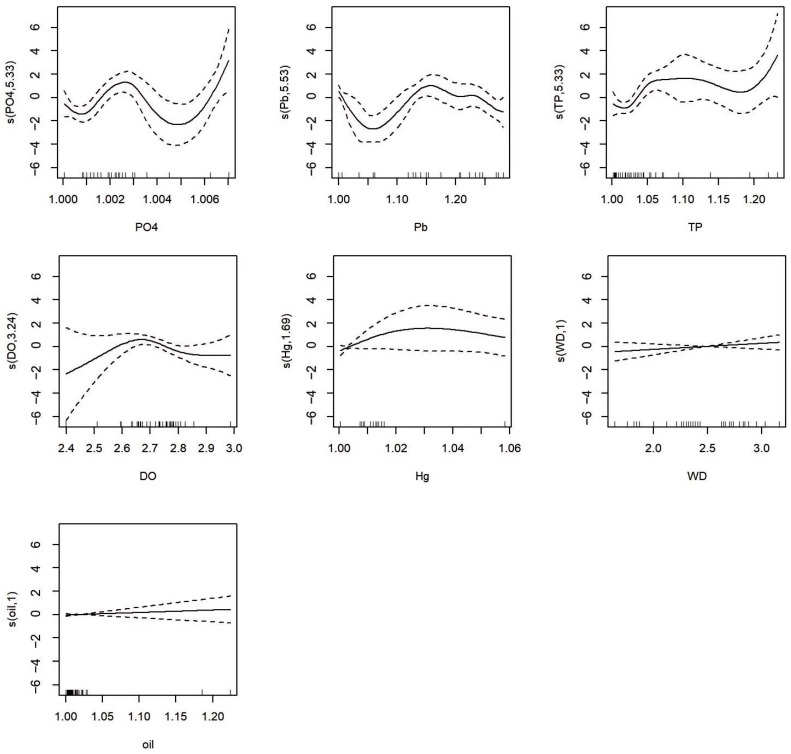
GAM plot for the Relationship between Phytoplankton Species Richness and Environmental Factors in the Coastal Waters of Beihai Silver Beach. The solid line represents the fitted values after smoothing, while the area between the dashed lines denotes 95% confidence intervals for the effect.

**Figure 8 biology-15-00034-f008:**
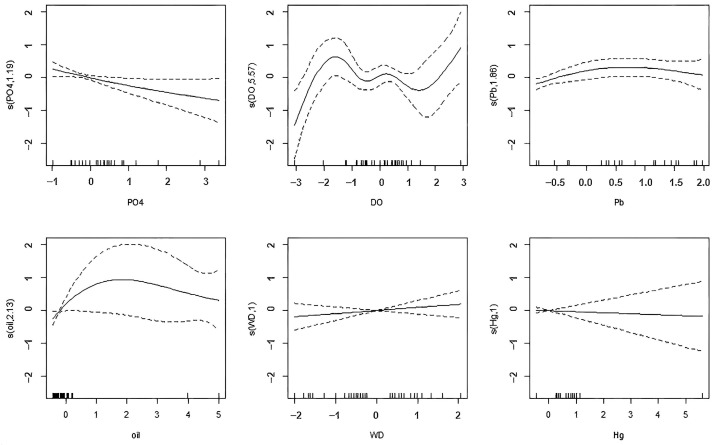
GAM plot for the Relationship between Phytoplankton Biodiversity Index and Environmental Factors in the Coastal Waters of Beihai Silver Beach. The solid line represents the fitted values after smoothing, while the area between the dashed lines denotes 95% confidence intervals for the effect.

**Table 1 biology-15-00034-t001:** Seasonal Variations in Environmental Factors in the Nearshore Waters of Beihai Silver Beach.

Environmental Factors	December 2020	September 2021	August 2022
WT (°C)	19.31 ± 0.09	30.4 ± 0.09	30.7 ± 0.13
WD (m)	4.38 ± 0.21	6.07 ± 0.45	5.49 ± 0.57
Sal	31.39 ± 0.03	26.06 ± 0.17	21.18 ± 1.40
DO (mg/L)	6.67 ± 0.03	6.02 ± 0.13	6.45 ± 0.16
pH	8.07 ± 0.01	8.27 ± 0.01	8.19 ± 0.02
SS (mg/L)	5.82 ± 0.34	5.82 ± 0.33	9.93 ± 0.45
Oil (mg/L)	0.08 ± 0.04	0.01 ± 0.002	0.02 ± 0.004
PO_4_^3−^ (mg/L)	0.001 ± 0.0008	0.006 ± 0.0006	0.003 ± 0.0006
SiO_3_^2−^ (mg/L)	0.29 ± 0.04	0.09 ± 0.01	0.21 ± 0.01
NO^2−^-N (mg/L)	0.0003 ± 0.0001	0.001 ± 0.0004	0.029 ± 0.005
NO^3−^-N (mg/L)	0.015 ± 0.003	0.005 ± 0.001	0.11 ± 0.02
NH^4+^-N (mg/L)	0.029 ± 0.01	0.008 ± 0.005	0.003 ± 0.001
COD (mg/L)	0.31 ± 0.09	1.92 ± 0.07	1.68 ± 0.11
TP (mg/L)	0.01 ± 0.001	0.19 ± 0.04	0.05 ± 0.007
TN (mg/L)	3.17 ± 0.34	0.03 ± 0.005	0.22 ± 0.02
Chl-a (mg/L)	1.87 ± 0.27	0.01 ± 0.001	3.06 ± 0.32
Cu (μg/L)	1.1 ± 0.00	1.39 ± 0.72	10.22 ± 7.20
Pb (μg/L)	0.03 ± 0.00	0.16 ± 0.17	0.42 ± 0.18
Zn (μg/L)	3.1 ± 0.00	3.83 ± 2.21	48.23 ± 15.10
Cd (μg/L)	0.01 ± 0.00	0.09 ± 0.02	0.01 ± 0.00
Hg (μg/L)	0.028 ± 0.03	0.007 ± 0.00	0.007 ± 0.00
As (μg/L)	0.79 ± 0.05	0.58 ± 0.11	0.53 ± 0.09
Cr (μg/L)	0.4 ± 0.00	0.45 ± 0.11	0.04 ± 0.00

Note: Values are presented as mean ± standard deviation (SD).

**Table 2 biology-15-00034-t002:** Dominant Species and Their Dominance of Phytoplankton.

Dominant Species	December 2020		September 2021		August 2022	
	Occurrence Frequency/%	Y	Occurrence Frequency/%	Y	Occurrence Frequency/%	Y
*Skeletonema costatum*	100.00	0.380	100.00	0.395	100.00	0.023
*Melosira sulcata*	60.00	0.067			100.00	0.028
*Thalassionema nitzschioides*	93.33	0.056				
*Chaetoceros lorenzianus*	93.33	0.055	100.00	0.174		
*Rhizosolenia imbricata*	93.33	0.050				
*Chaetoceros affinis*	86.67	0.041	100.00	0.181		
*Pseudo-nitzschia pungens*	73.33	0.025				
*Bacteriastrum hyalinum*			100.00	0.120	100.00	0.133
*Thalassiothrix frauenfeldii*			100.00	0.023		
*Hemidiscus cuneiformis*					100.00	0.270
*Bacteriastrum varians*					100.00	0.112
*Nitzschia longissima*					100.00	0.042
*Chaetoceros pelagicus*					100.00	0.035
*Nitzschia sublanceolata*					100.00	0.029
*Peridinium pentagonum*					100.00	0.023

Note: The abundance data for chain-forming species are based on counts of individual cells.

**Table 3 biology-15-00034-t003:** The fitness of the model.

Model Fitness	S	H′
	Values	Values
R^2^	0.91	0.436
Generalized Cross-Validation (GCV)	0.90	0.354

**Table 4 biology-15-00034-t004:** The optimal model parameters for the relationship between phytoplankton species richness and environmental factors.

Variables	Effective Degrees of Freedom (edf)	Reference Degree of Freedom (Ref.df)	F Value	*p* Value
s (PO_4_^3−^)	5.333	6.077	5.878	9.080 × 10^−4^ ***
s (Pb)	5.534	6.198	4.782	3.553 × 10^−3^ **
s (TP)	5.326	6.086	4.193	5.939 × 10^−3^ **
s (DO)	3.243	3.947	2.767	0.062
s (Hg)	1.690	1.932	1.344	0.276
s (WD)	1.000	1.000	1.171	0.291
s (oil)	1.000	1.000	0.567	0.459

**, *p* < 0.01; ***, *p* < 0.001.

**Table 5 biology-15-00034-t005:** The optimal model parameters for the relationship between phytoplankton biodiversity index and environmental factors.

Variables	Effective Degrees of Freedom (edf)	Reference Degree of Freedom (Ref.df)	F Value	*p* Value
s (PO_4_^3−^)	1.195	1.350	4.445	0.036 *
s (DO)	5.574	6.567	2.732	0.027 *
s (Pb)	1.862	2.246	2.742	0.077
s (oil)	2.133	2.586	2.149	0.132
s (WD)	1.000	1.000	0.867	0.359
s (Hg)	1.000	1.000	0.112	0.740

*, *p* < 0.05.

## Data Availability

The data supporting this study’s findings are available from the corresponding authors upon reasonable request.

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
