# Peer review of "Nonlinear Responses of Phytoplankton Communities to Environmental Drivers in a Tourist-Impacted Coastal Zone: A GAMs-Based Study of Beihai Silver Beach"

_biology, 2025, doi:10.3390/biology15010034_

Round 1
Reviewer 1 Report
Comments and Suggestions for Authors
Summary: Dang et al. analyze phytoplankton and environmental data from three sampling campaigns (December 2020, September 2021, and August 2022) at Beihai Beach in Southern China. The paper is well-written, straightforward, and relies on established methods. My biggest concern is with the rigor of the work, as several arguments are made without sufficient evidence to support them.
Major concerns:
- The sampling was conducted with a net of mesh size 0.077mm, or about 77um (Ln 98). Given this study’s focus on microplankton, this means that many species, including non-chain forming diatoms, will not be sufficiently sampled. Please explicitly explain what effects this sampling approach has on the results. Maybe the dinoflagellates were too small to be captured. Please also state how many samples were collected for each season and how often (daily/weekly/opportunistic).
- Please explain the following patterns and incorporate into the discussion:
- The water temperature in the Autumn 2021 and Summer 2022 is essentially the same.
- Autumn 2021 has much lower silica and nitrate concentrations than the other seasons, likely indicative of a post-diatom bloom environment.
- The chlorophyll value of Autumn 2021 is very low (0.01mg/L), but the phytoplankton richness is similar to Summer 2022 (Ln 193). Interestingly, the phytoplankton abundance reported is 1000 times greater than the phytoplankton abundance in Winter 2020 (Ln 221-234).
- The copper and zinc ion concentrations for Summer 2022 is an order of magnitude greater than the other seasons, whereas chromium is an order of magnitude smaller.
- Most of the dominant taxa reported are chain-forming. Were single variants of these species observed at all?
Minor comments:
Ln 209: Change ‘dminant’ to dominant.
Figure 4: These panels need to be bigger. The legend cannot be read.
Ln 256: Capitalize ‘diversity’
Ln 368-372: This statement does not seem correct.
Author Response
|
Response to Reviewer 1 Comments Summary: Dang et al. analyze phytoplankton and environmental data from three sampling campaigns (December 2020, September 2021, and August 2022) at Beihai Beach in Southern China. The paper is well-written, straightforward, and relies on established methods. My biggest concern is with the rigor of the work, as several arguments are made without sufficient evidence to support them. |
||
|
1. Summary |
|
|
|
We sincerely thank Reviewer for their constructive and insightful evaluation of our manuscript. We greatly appreciate their positive assessment regarding its clarity and methodology, as well as their thoughtful identification of areas where the scientific rigor could be strengthened. Addressing these points has been invaluable in improving the overall quality of our work. We are particularly grateful for the specific, detailed comments provided, which have guided a comprehensive revision of the manuscript. Below, we provide our point-by-point responses to each comment and detail the corresponding revisions made.
|
||
|
1. Point-by-point response to Comments and Suggestions for Authors |
||
Comment 1: The sampling was conducted with a net of mesh size 0.077mm, or about 77um (Ln 98). Given this study’s focus on microplankton, this means that many species, including non-chain forming diatoms, will not be sufficiently sampled. Please explicitly explain what effects this sampling approach has on the results. Maybe the dinoflagellates were too small to be captured. Please also state how many samples were collected for each season and how often (daily/weekly/opportunistic).
Response 1: We sincerely thank the reviewer for raising this critical point regarding methodological limitations. The reviewer is absolutely correct to note that a 77 μm mesh net preferentially captures larger microplankton (e.g., chain-forming diatoms, large dinoflagellates) and undersamples smaller nanoplankton and picoplankton, including single-celled diatoms and many small dinoflagellates. This inherent bias means our reported community composition and species richness likely underestimate the contributions of smaller-sized fractions.
Actions Taken: We have added explicit statements in the Methods section (Lines 102-107) acknowledging this limitation. Furthermore, we have expanded the Discussion section (section 4.1) to discuss how this sampling approach shapes our findings, specifically cautioning that our conclusions pertain primarily to the "net-phytoplankton" community. We also now suggest that future studies integrate water bottle sampling for a more complete assessment of the entire microplankton size spectrum.
Clarification on Sampling Details: We apologize for the omission of these details. The sampling was conducted during intensive, short-term campaigns. For each seasonal campaign (Dec-2020, Sep-2021, Aug-2022), surface water samples were collected once per season across all 15 stations, with the entire campaign completed within two consecutive days at each station, yielding a total of 15 samples per season . This clarification has been added to the Methods section (Lines 95-98).
Revise the content: Surface water samples were obtained from 15 stations within the nearshore environment of Silver Beach (Figure 1), with one sample collected per station for a total of 15 samples. The sampling design focused on the central tourism district and the Fengjia River estuary to encompass regions subject to substantial anthropogenic pressure. Seasonal sampling was conducted once per quarter in December 2020, September 2021, and August 2022, with each campaign completed within two consecutive days, representing characteristic hydrological conditions during winter, autumn, and summer, respectively.
Comment 2: Please explain the following patterns and incorporate into the discussion:
- The water temperature in the Autumn 2021 and Summer 2022 is essentially the same.
(2)Autumn 2021 has much lower silica and nitrate concentrations than the other seasons, likely indicative of a post-diatom bloom environment.
(3)The chlorophyll value of Autumn 2021 is very low (0.01mg/L), but the phytoplankton richness is similar to Summer 2022 (Ln 193). Interestingly, the phytoplankton abundance reported is 1000 times greater than the phytoplankton abundance in Winter 2020 (Ln 221-234).
(4)The copper and zinc ion concentrations for Summer 2022 is an order of magnitude greater than the other seasons, whereas chromium is an order of magnitude smaller.
Response 2: Thank you for these insightful observations regarding the distinct seasonal patterns in our environmental and phytoplankton data. We fully agree that these patterns are critical for interpreting the ecosystem dynamics at Beihai Silver Beach and have incorporated detailed explanations for each point into the revised discussion section (Section 4.1 and section 4.2).
Revise the content: (1) Regarding the similar water temperatures in Autumn 2021 and Summer 2022
We acknowledge this point and have added a sentence to highlight that while temperatures were comparable, the observed community structures (e.g., much lower diversity in autumn 2021) differed significantly. This underscores our conclusion that factors other than temperature—particularly nutrient dynamics and pollutant loads—were the primary drivers of the distinct ecological states observed in these two seasons.
- Regarding the low silica and nitrate concentrations in Autumn 2021
We agree that this is a key indicator. We have explicitly linked the extremely low silicate and nitrate concentrations recorded in autumn 2021 to the occurrence of a major diatom bloom event that preceded our sampling. This interpretation is now integrated into the paragraph discussing the Skeletonema costatum bloom, stating that these depleted nutrient levels are characteristic of a post-diatom bloom environment and support the identification of autumn 2021 as a post-bloom period.
- Regarding the decoupling of low chlorophyll-a, high species richness, and extreme cell abundance in Autumn 2021
This was a particularly intriguing finding, and we thank the reviewer for prompting a deeper discussion. We have added a dedicated paragraph to Section 4.1 to address this specific pattern. We propose and discuss several plausible explanations for this decoupling: The dominance of small-sized diatoms like S. costatum, which have a lower chlorophyll content per cell. The potential prevalence of senescent or physiologically stressed cells following the bloom peak, which would be counted in microscopic abundance but contribute little to photosynthetic biomass. The methodological consideration that our chlorophyll-a measurement integrates the entire water column and size spectrum, potentially including pico- and nanophytoplankton not captured by our 77 µm net, whereas our abundance data is based on the net-collected fraction.
(4) Regarding the anomalous trace metal concentrations in Summer 2022
We have added a new paragraph at the end of Section 4.2 to discuss this striking pattern. We highlight that the order-of-magnitude differences in Cu, Zn, and Cr concentrations point to a specific, transient anthropogenic source during that period (e.g., anti-fouling paints, localized runoff, or industrial discharge). We further discuss the potential ecological implications, noting that elevated Cu and Zn could act as additional stressors on the phytoplankton community, potentially contributing to the unique community structure observed in summer 2022 and emphasizing the variable nature of pollutant pressures in this coastal zone. By incorporating these points, we believe the revised discussion now provides a more rigorous, nuanced, and comprehensive interpretation of the complex spatiotemporal dynamics observed in our study. We thank the reviewer again for these valuable comments.
Comment 3: Most of the dominant taxa reported are chain-forming. Were single variants of these species observed at all?
Response 3: Thank you for raising this important methodological point. We appreciate the opportunity to clarify our counting procedure regarding colonial and chain-forming taxa. In our study, all phytoplankton enumeration was performed at the cellular level. For chain-forming or colonial species (such as Skeletonema costatum, Chaetoceros spp., and Bacteriastrum spp.), the reported abundances represent counts of individual cells, not colonies or chains. Therefore, single cells of these species were indeed observed and counted when they occurred as solitary units. However, it is important to note that in our samples, these taxa were predominantly present in their characteristic chain or colonial form. The counting of individual cells within chains is the standard practice in phytoplankton quantification, as it allows for a more accurate and ecologically relevant comparison of biomass and population size across different morphological groups.
Revise the content: To prevent any ambiguity, we have now explicitly stated this counting protocol in the revised Section 2.2. Sampling and Analysis:"...species identification and enumeration were conducted under an optical microscope. For colonial or chain-forming species, counts represent the number of individual cells."
Additionally, a clarifying note has been added to the caption of Table 2 (Dominant Species) in the revised manuscript: "Note: The abundance data for chain-forming species are based on counts of individual cells." We believe these clarifications address your concern and ensure the transparency of our methodology. Thank you again for your careful review.
Comment 4: Ln 209: Change ‘dminant’ to dominant.
Response 4: Thank you for this editorial correction. We have capitalized "diversity" as required in the relevant context.
Comment 5: Capitalize ‘diversity’.
Response 5: Thank you for this editorial correction. We have capitalized "diversity" as required in the relevant context.
Comment 6: Ln 368-372: This statement does not seem correct.
Response 6: We sincerely apologize for the lack of clarity in the original sentence on lines 215-217. The wording was indeed confusing. We have revised this section to improve its readability and precision.
Reviewer 2 Report
Comments and Suggestions for Authors
The study presents multi-year data on phytoplankton dynamics in a coastal ecosystem, employs an advanced statistical model (GAMs), and offers insights relevant for environmental management. The work is generally well-structured. However, there are several issues, ranging from statistical reporting to textual clarity and data interpretation, that need to be addressed before the manuscript can be considered for publication.
-The use of adjusted R-squared (R²adj) in the abstract and results to describe a single model's fit is not standard. R² is the appropriate metric to report the proportion of variance explained by that specific model. R²adj is primarily useful for comparing models with different numbers of predictors. Please replace all instances of reported R²adj with R² for clarity.
-The keyword list contains redundant terms (e.g., "phytoplankton" and "phytoplankton community"). Moreover, I did not find any morphological traits in the main text.
-Line 132: “the i species” should be changed as "the ith species" or "Species i".
-Lines 137-138: The definitions for n_i and N are indeed duplicated. Remove the redundant sentence.
-Lines 143-144: The term "parameters" is incorrect here. These are your response variables​ (e.g., species richness, diversity indices). The notation "Nppt" is confusing and inconsistent. The text previously uses S for species richness. Use consistent symbols. Also, clarify what "ppt" stands for if it is an abbreviation used elsewhere.
-Table 1: The table note must explicitly state what the values after "±" represent (e.g., "Values are mean ± standard deviation (SD)" or "± standard error (SE)").
-Lines 201-202: The results about intercept is not important at all. Readers would not care about it.
-The statement "summer > autumn" appears to contradict the visual data in Figure 2B (Autumn 2021), which seems to show higher values than Figure 2C (Summer). Please re-examine the statistical analysis and the figure. The conclusion must align with the presented data.
-Line 209: Typo.
-Line 211: "spring 2022" is mentioned, but the preceding text refers to "August." August is considered a summer month.
-Line 214: The species ”Thalassiothrix frauenfeldii” should be removed from the list, as it only appeared once according to Table 2.
-Lines 215-217: I cannot understand what does this sentence state.
-Line 256: "Diversity" should be capitalized. More importantly, all the results in this section are descriptive. Some statistical models should be employed to support the conclusion.
-Lines 286-290: The results about intercept is not important at all. Readers would not care about it. I suggest removing the sentences.
Author Response
|
The study presents multi-year data on phytoplankton dynamics in a coastal ecosystem, employs an advanced statistical model (GAMs), and offers insights relevant for environmental management. The work is generally well-structured. However, there are several issues, ranging from statistical reporting to textual clarity and data interpretation, that need to be addressed before the manuscript can be considered for publication. 1. Summary |
|
We sincerely thank the reviewer for their thorough and constructive evaluation of our manuscript. Their insightful comments and suggestions have been invaluable in improving the clarity, accuracy, and scientific rigor of our work. We have carefully considered each point and have revised the manuscript accordingly. Please find our detailed point-by-point responses below.
|
|
2. Point-by-point response to Comments and Suggestions for Authors |
|
Comments 1: The use of adjusted R-squared (R²adj) in the abstract and results to describe a single model's fit is not standard. R² is the appropriate metric to report the proportion of variance explained by that specific model. R²adj is primarily useful for comparing models with different numbers of predictors. Please replace all instances of reported R²adj with R² for clarity. |
|
Response 1: We thank the reviewer for this important clarification regarding statistical reporting. The reviewer is correct. We have now replaced all instances of R²adj with R² throughout the manuscript (in the Abstract, Results, and Figure captions as applicable) to accurately report the proportion of variance explained by each individual GAM.
|
|
Comments 2: The keyword list contains redundant terms (e.g., "phytoplankton" and "phytoplankton community"). Moreover, I did not find any morphological traits in the main text. |
|
Response 2: We appreciate the reviewer's attention to detail. We have revised the keyword list to eliminate redundancy, removing "phytoplankton " as suggested. Revise the content: Phytoplankton community; Beihai Silver Beach; GAMs; Environmental factor; Species richness; Diversity index.
Comments 3: Line 132: “the i species” should be changed as "the ith species" or "Species i". Response 3:Thank you for pointing this out. We have corrected the text to "the ith species" as recommended (line 144).
Comments 4: Lines 137-138: The definitions for n_i and N are indeed duplicated. Remove the redundant sentence. Response 4: We appreciate the reviewer's correction. To clarify, the initial definitions for náµ¢ (the abundance of the ith species) and N (the total abundance of all species) were indeed present but expressed redundantly. As suggested, the duplicate clarifying sentence has been removed. The text now correctly states: "Where náµ¢ represents the abundance of the ith species; N denotes the total abundance of all species in the study area."(line 144-145)
Comments 5: Lines 143-144: The term "parameters" is incorrect here. These are your response variables (e.g., species richness, diversity indices). The notation "Nppt" is confusing and inconsistent. The text previously uses S for species richness. Use consistent symbols. Also, clarify what "ppt" stands for if it is an abbreviation used elsewhere. Response 5: Thank you for pointing this out. We agree with this comment. Species richness, diversity indices are truly response variables. Therefore, we have revised the sentence as follows: “Generalized Additive Models (GAMs) were employed to conduct an in-depth analysis of the nonlinear relationships between multiple environmental factors and phytoplankton′ s response variables (S and H′). ”. Furthermore, to keep all the symbols consistent, we changed Nppt to S, and Hppt to H’.
Comments 6: Table 1: The table note must explicitly state what the values after "±" represent (e.g., "Values are mean ± standard deviation (SD)" or "± standard error (SE)"). Response 6: Thank you for this necessary clarification. We have updated the note for Table 1 to explicitly state: "Values are presented as mean ± standard deviation (SD)."(line 198)
Comments 7: Lines 201-202: The results about intercept is not important at all. Readers would not care about it. Response 7: We agree with the reviewer that the intercept values are not a focal point for interpretation in the context of our GAMs. As suggested, We have removed the previous Table 3 (The parameter coefficient of the intercept) and some related statements in the main text.
Comments 8:The statement "summer > autumn" appears to contradict the visual data in Figure 2B (Autumn 2021), which seems to show higher values than Figure 2C (Summer). Please re-examine the statistical analysis and the figure. The conclusion must align with the presented data. Response 8: We sincerely thank the reviewer for their careful observation. The reviewer is correct that the statement "summer > autumn" does not align with the visual trend shown in Figure 2. Upon re-checking, this was indeed an error in our textual description. The correct and statistically significant seasonal comparison should be autumn > summer. We have corrected this statement in the manuscript and performed a thorough review of the entire text to ensure that all descriptions of seasonal trends are consistent with the statistical results and the data presented in the figures. We apologize for this oversight.
Comments 9: Line 209: Typo. Response 9: Thank you. The typographical error on line 209 has been corrected.
Comments 10: Line 211: "spring 2022" is mentioned, but the preceding text refers to "August." August is considered a summer month. Response 10: We appreciate the reviewer catching this inconsistency. This was an error in the text. We have corrected line 211 to refer to "summer 2022" to be consistent with the August sampling date.
Comments 11: Line 214: The species ”Thalassiothrix frauenfeldii” should be removed from the list, as it only appeared once according to Table 2. Response 11: Thank you for this careful review. The reviewer is correct. We have removed "Thalassiothrix frauenfeldii" from the list of dominant species in the text , as its single occurrence does not meet our stated criteria for dominance.
Comments 12: Lines 215-217: I cannot understand what does this sentence state. Response 12: We sincerely apologize for the lack of clarity in the original sentence on lines 215-217. The wording was indeed confusing. We have revised this section to improve its readability and precision. The modified text now reads: "The dominant phytoplankton species comprised 7 taxa in winter 2020, 5 in autumn 2021, and 9 in summer 2022. Diatoms predominated across all surveys. Skeletonema costatum was identified as a dominant species in all three sampling campaigns, with dominance values of 0.380 (winter 2020), 0.395 (autumn 2021), and 0.023 (summer 2022). Species such as Chaetoceros lorenzianus, Chaetoceros affinis, and Bacteriastrum hyalinum occurred as dominants in two different seasons. In contrast, only one dinoflagellate species, Peridinium pentagonum, was recorded as a dominant species, primarily during the summer 2022 survey, with a dominance value of 0.023 (Table 2)."
Comments 13: Line 256: "Diversity" should be capitalized. More importantly, all the results in this section are descriptive. Some statistical models should be employed to support the conclusion. Response 13: We sincerely thank the reviewer for this critical suggestion. We have revised Section 3.2.4 “Diversity index” (the heading has been capitalized as suggested) to incorporate rigorous statistical analysis that supports our descriptive findings. Specifically, following the descriptive presentation of the seasonal ranges and mean values for the Shannon–Wiener index (H′) and Pielou’s evenness index (J), we have now added a new paragraph of statistical inference. Since the data did not meet the normality assumption required for parametric tests, we employed the non-parametric Kruskal–Wallis rank-sum test to formally assess whether the observed seasonal differences were statistically significant. The results confirm that the overall seasonal differences are highly statistically significant for both indices. This quantitative analysis strengthens our conclusion that phytoplankton community diversity and evenness varied markedly across seasons, with Autumn 2021 showing significantly lower values. We believe this addition addresses the reviewer’s concern by moving beyond mere description to provide statistical evidence for our claims.
Comments 14: Lines 286-290: The results about intercept is not important at all. Readers would not care about it. I suggest removing the sentences. Response 14: We concur with the reviewer. Following the same principle as for comment 7, we have removed the sentences reporting the intercept values from lines 286-290 in the Discussion/Conclusion section. |
Round 2
Reviewer 1 Report
Comments and Suggestions for Authors
Thank you for the revisions. Please carefully check for minor errors, such as floating quotation marks (Ln 104), inconsistent italicization of species names (e.g. Ln 455), and repetition (Ln 456).
Author Response
|
Response to Reviewer 1 Comments |
||
|
|
|
|
|
We sincerely extend our gratitude to the Reviewer for their thorough and attentive review of our revised manuscript. We greatly appreciate the time and effort dedicated to providing further constructive feedback. We have carefully considered all the comments and suggestions raised in this round of review and have fully incorporated them into the manuscript. Below, we provide point-by-point responses to each comment and detail the corresponding revisions that have been made.
|
||
|
Point-by-point response to Comments and Suggestions for Authors |
||
|
Comment 1: floating quotation marks (Ln 104) Response 1: We thank the Reviewer for identifying the formatting issue. The floating quotation marks on Line 104 have been corrected and are now properly integrated into the text.
Comment 2: inconsistent italicization of species names Response 2: We appreciate the Reviewer's attention to taxonomic formatting. All species names throughout the manuscript have been reviewed and italicized consistently according to standard convention.
Comment 3: repetition (Ln 456). Response 3: We sincerely apologize for this oversight and the inconvenience it may have caused. Upon reviewing Line 456, we have been unable to identify the specific repetitive phrasing the Reviewer is referring to. To ensure we address it accurately, could you kindly provide a brief indication of the specific content in question? We would be very grateful for this clarification so that we can make the appropriate revision. Thank you very much for your assistance.
Once again, we are grateful for your valuable insights, which have helped us further improve the quality and accuracy of our manuscript. |
||
Reviewer 2 Report
Comments and Suggestions for Authors
This revision has improved significantly and addressed most of my concerns. There are still a few minor issues:
-Lns 144–145: The explanation of n and N has already been mentioned earlier (Lns 138-140), so it should not be repeated here.
-Ln 167: It should not be “adjusted coefficient of determination”. Please remove “adjusted”. Also, “deviance explained” is essentially R², so it should not be stated again redundantly.
-Lns 256–262: The figure appears to show the spatial distribution obtained through simple spatial interpolation of data points, without considering the influence of land-sea distribution on the interpolation. Instead, land areas are simply removed from the interpolation results. The interpolated results could be misleading. I recommend displaying abundance only through the color or size of the sampling points, without performing interpolation.
-Ln 307: It is suggested to change “Deviance explained” in the table to R².
Author Response
|
Response to Reviewer 2 Comments |
|
Thank you very much for your further careful review and constructive suggestions. We appreciate your positive feedback and agree with the remaining points raised. We have addressed each comment accordingly in the revised manuscript. |
|
|
|
2. Point-by-point response to Comments and Suggestions for Authors |
|
Comments 1: The explanation of n and N has already been mentioned earlier (Lns 138-140), so it should not be repeated here. |
|
Response 1: We sincerely thank you for this observation. You are correct that the explanation of n and N is redundant here, as it was already clearly defined earlier. We have removed the repeated explanation to improve the conciseness of the text.
|
|
Comments 2: It should not be “adjusted coefficient of determination”. Please remove “adjusted”. Also, “deviance explained” is essentially R², so it should not be stated again redundantly. |
|
Response 2: Thank you for pointing out these inaccuracies. We agree with your suggestions. We have removed "adjusted" and now refer to it simply as the "coefficient of determination (R²)". We have also removed the redundant phrase "deviance explained" to avoid repetition.
Comments 3: The figure appears to show the spatial distribution obtained through simple spatial interpolation of data points, without considering the influence of land-sea distribution on the interpolation. Instead, land areas are simply removed from the interpolation results. The interpolated results could be misleading. I recommend displaying abundance only through the color or size of the sampling points, without performing interpolation. Response 3: We appreciate this critical comment regarding the spatial interpolation. We agree that the previous interpolation method, which did not account for the land-sea distribution and simply masked land areas, could be misleading. Following your recommendation, we have revised the relevant figure. The updated figure now displays abundance data solely through the color and/or size of the actual sampling points, without any spatial interpolation. The corresponding description in the text (Lns 256–262) has been updated to accurately reflect this change in methodology.
Comments 4: It is suggested to change “Deviance explained” in the table to R². Response 4: Thank you for the suggestion. We have changed "Deviance explained" to "R²" in the table as requested for clarity and consistency with standard terminology.
Once again, we are grateful for your valuable insights, which have helped us further improve the quality and accuracy of our manuscript.
|